# Effect of Breeding Techniques and Prolonged Post Dry Aging Maturation Process on Biomolecule Levels in Raw Buffalo Meat

**DOI:** 10.3390/vetsci8040066

**Published:** 2021-04-20

**Authors:** Angela Salzano, Alessio Cotticelli, Raffaele Marrone, Michael J. D’Occhio, Nunzia D’Onofrio, Gianluca Neglia, Rosa Luisa Ambrosio, Maria Luisa Balestrieri, Giuseppe Campanile

**Affiliations:** 1Department of Veterinary Medicine and Animal Production, University of Naples “Federico II”, 80137 Naples, Italy; angela.salzano@unina.it (A.S.); alessio.cotticelli@unina.it (A.C.); raffaele.marrone@unina.it (R.M.); neglia@unina.it (G.N.); rosaluisa.ambrosio@unina.it (R.L.A.); giucampa@unina.it (G.C.); 2School of Life and Environmental Sciences, Faculty of Science, The University of Sydney, Sydney, NSW 2000, Australia; michael.docchio@sydney.edu.au; 3Department of Precision Medicine, University of Campania “Luigi Vanvitelli”, 80128 Naples, Italy; marialuisa.balestrieri@unicampania.it

**Keywords:** buffalo meat, functional molecules, meat quality, post dry aging

## Abstract

Recently, several concerns have been expressed on red meat quality and consumption. The aims of this study were to evaluate the influence of different breeding techniques and a prolonged post dry aging (PDA) maturation process on biomolecules level in raw buffalo meat. In the first experiment, two groups of animals were maintained with different space availability (15 vs. 10 m^2^/animal) for 90 days and biomolecules content was evaluated. In experiment 2, two diets (with or without ryegrass green forage) were used to assess the concentration of these biomolecules. Finally, in experiment 3, the meat of the animals that showed the highest content of biomolecules was chosen to assess the influence of the PDA maturation process. Buffaloes reared at 15 m^2^ showed a significantly (*p* < 0.05) higher content of the considered biomolecules compared with their counterparts. Similarly, buffaloes fed green forage showed higher content of biomolecules (*p* < 0.05) compared with the control group. The meat of the animals bred at 15 m^2^ and fed green forage showed a significant (*p* < 0.01) increase of biomolecules content during the PDA maturation process up to 60 days without influence microbiological profile in terms of total aerobic bacterial counts, yeasts, and molds. In conclusion, breeding techniques and PDA maturation system could enhance biomolecules levels in terms of quality, without affect health standards.

## 1. Introduction

Nowadays, the future of livestock production is based on the “One health” approach. Indeed, environmental and economic sustainability, combined with animal welfare, is able to guarantee and enhance the quality of animal products. This is especially important for some kinds of products, such as red meat, which has been an important constituent of the human diet throughout human evolution. Indeed, red meat provides a rich source of high biological value proteins and essential micronutrients when it is part of a healthy, varied diet [1]. Red meat also has an active role on immunity [2,3] and enhances cognitive abilities in children [4], as it is a good source of zinc, iron, selenium, calcium, phosphorus, and lipids, followed by vitamin A- and B-complex vitamins [5,6].

Among different types of red meat, buffalo meat has a similar composition to bovine counterpart at the same age [7], but has lower fat, cholesterol, and calories content [8]. Indeed, buffalo meat seems to be associated with some beneficial effects on cardiovascular risk profile, including lower carotid atherosclerotic burden and susceptibility to oxidative stress [9]. However, in recent years, meat has been demonized by the final consumer after numerous sanitary problems (e.g., mad cow disease, avian influenza) and attracted much debate regarding its impact on health. Prospective cohort and epidemiological studies pointed out the potential adverse health effects of fresh and transformed red meats consumption on major chronic diseases, such as cardiovascular diseases and cancer [10,11,12]. This relationship is partially explained by the excessive amount and quality of the fat found in red meat. In particular, meat deriving from ruminants can be rich in saturated fatty acids and cholesterol [13,14,15]. The lipid hypothesis has had broad and huge consequences; not only in consumers’ perspective, but also on the way food is produced on-farm. Indeed, in the last years, consumers focused their preferences towards grass-fed animals that tend to have lower fat content and higher precursors for vitamin A and E [16,17]. Moreover, in grass-fed animals, there is a higher presence of antioxidants such as glutathione and super oxide dismutase compared with their grain-fed counterparts [18,19].

Nowadays, more attention is shifted towards the nutraceutical profile of animal products. Several functional molecules, essential in human health, such as betaine (δ-Valerobetaine, γ-Butyrobetaine, Glycine betaine), L-carnitine, and its precursors, have been recently found in ruminant milk and meat, particularly in buffaloes [20]. Among these biomolecules, δ-Valerobetaine showed both antioxidant and anti-inflammatory activities and is able to counteract endothelial oxidative stress and inflammation [21]. In addition, the anticancer effect of δ-Valerobetaine, displayed in HT-29 and LoVo human colon adenocarcinoma colon cancer cells [22] and in head and neck squamous cell carcinomas [23], indicated it as a novel dietary compound with health-promoting activities. Of interest, a synergistic action in inhibiting proliferation and inducing apoptosis in oral squamous Cal 27 cells has been observed after the combination of δ-Valerobetaine and γ-Butyrobetaine [23].

As biomolecules confer high nutritional value and health-promoting properties, it is of utmost importance to maintain their adequate levels in buffalo meat. It has already been reported that management techniques could influence the amount of these biomolecules in buffalo milk [24,25], but nothing has been done on buffalo meat yet. It has been shown that enhancing animal welfare led to an increase of these biomolecules in both milk and dairy products in buffaloes [24]. Moreover, a green forage-based diet has improved the concentration of these functional molecules in milk and the antioxidant activity measured in both milk and blood [25]. Based on these results, it could also be interesting to assess the biomolecules content during a post dry aging (PDA) maturation process. It has already been demonstrated that a combined used of green forages and a PDA maturation period could improve both tenderness and colour in buffalo meat [26], but, until now, nothing has been done on PDA and functional molecules content. Moreover, in the meat industry, it is also important to follow the product from farm to fork to ensure that these biomolecules are not lost during meat transformation (e.g., during maturation processes or cooking). Hence, the aims of this study were to understand (1) whether production system could affect biomolecules content in buffalo meat and (2) if a prolonged PDA maturation process in a defined system (Maturmeat^®^) could influence the amount of biomolecules in raw buffalo meat.

## 2. Materials and Methods

### 2.1. Experimental Design

In order to understand if the production system could have a role in biomolecules’ production in buffalo meat, two different experiments were performed. The first experiment was performed in order to see whether the availability of space could influence the amount of functional molecules in buffalo meat and the biomolecules’ content was measured in two groups of animals reared in different space allocation. The space availability (15 m^2^/animal) of the group with a higher production of biomolecules was chosen and the other two groups of animals for experiment two were maintained with that animal density. The second experiment was performed to understand if the diet could modify the content of functional molecules in meat. Two different diets were created, with and without the use of green forage. Animals fed with green forage had higher biomolecules content in meat. The meat of the animals bred with higher space availability and fed with green forage was chosen to see if a prolonged maturation process could also influence the amount of biomolecules in raw buffalo meat.

### 2.2. Effect of Availability of Space on Functional Molecules in Meat

#### Animals and Diet

The study was carried out in a commercial buffalo dairy (Verdi Praterie, Kr, Italy) on 16 Italian Mediterranean buffalo bulls. The animals involved in the trial had an average age of 595 ± 0.7 days and weight of 455 ± 7.0 kg. Buffaloes were kept in pens with a concrete floor and randomly divided into two groups: Group S−, space allocation of 10 m^2^/buffalo; Group S+, space allocation of 15 m^2^/buffalo. Animals were fed twice daily: in the morning (06:00) and in the afternoon (17:00). Every day, the amount of refusals was removed, weighed, and used to calculate dry matter intake (DMI). Energy values (MFU = 1700 kcal net energy for lactation (NEL) were calculated using equations provided by the Institute National de Recherche Agronomique [27] and the Nutrient Requirement of Dairy Cattle [28]. The study lasted three months (fattening period). The amount and the composition of the diet administered to both groups are reported in Table 1.

### 2.3. Effect of Feeding on Functional Molecules in Meat

Taking into account the results from experiment 1, the study was carried out in a commercial buffalo farm (Verdi Praterie, Kr, Italy) on 16 Italian Mediterranean buffalo bulls. The animals involved in the trial had an average age of 581 ± 0.4 days and weight of 460 ± 3.5 kg. Buffaloes kept in pens with concrete floor with a space allocation of 15 m^2^/buffalo (S+) were randomly divided into two groups: with (S+/F; n = 8) or without (S+/D; n = 8) rye grass (*lolium multiflorum*) inclusion. The two diets were isonitrogenous and isoenergetic and differed only in the inclusion of green feed. Animals were fed twice daily: in the morning (06:00) and in the afternoon (17:00). Every day, the amount of refusals was removed, weighed, and used to calculate dry matter intake (DMI). Energy values (MFU = 1700 kcal net energy for lactation (NEL) were calculated using equations provided by the Institute National de Recherche Agronomique [27] and the Nutrient Requirement of Dairy Cattle [28]. The study lasted three months (fattening period). The amount and the composition of the diets are reported in Table 2.

### 2.4. Slaughter Procedures

For both experiments, buffaloes were slaughtered at a European Community-approved abattoir in compliance with European Community laws on Animal Welfare in transport (1/2005EC) and the European Community regulation on Animal Welfare for slaughter of commercial animals (1099/2009EC). Hot carcasses were weighed and graded for conformation (1: poor to 5: excellent) and fatness (1: low to 5: very high) using European grading systems for bovine carcasses (S.E.U.R.O.P.; EC 1208/1981). The carcasses were dressed without any electrical inputs and chilled at 4 °C. After 24 h chilling, the left side was separated into commercial joints and dissected into muscle, bone, subcutaneous fat, intermuscular fat, and tissues [29,30]. At the same time, the muscle/bone (M/B) and muscle/fat (M/F) percentage (%) were determined.

### 2.5. Post Dry Ageing (PDA)

After five days post-mortem ageing, the S+/F group underwent a period of PDA into the MaturMeat^®^ (European Patented Device and meat Dry Aging Method with safe and controlled Ph—n. EP 2769276B1). A section of Longissimus muscle (for a total of seven vertebrae, starting from the sixth to the last thoracic vertebrae) was removed from the carcasses and placed into the MaturMeat^®^. The Maturmeat^®^ consists of a monitored refrigeration device in which operators can customize ageing conditions, such as temperature and relative humidity. For this experiment, temperature and humidity were set to 2 °C and 78%, respectively. Buffalo meat cuts were placed on steel grids and monitored constantly to avoid mold growing or any external font of variability. Samples were weighed and each length dimension was registered et each experimental time of PDA in MaturMeat^®^ (T0 = 0 d aging; T1 = 30 d post aging; T2 = 60 d post aging), in order to control the data during the time and to measure weight loss during periods.

### 2.6. Physical-Chemical Analyses

The pH measurement was done at the slaughter and at 30 and 60 days of PDA with a digital pH meter (Crison-Micro TT 2022, Crison Instruments, Barcelona, Spain). At each measurement, the pH meter was previously automatically calibrated for muscle temperature using standard solutions with 4 and 7 pH values (Crison Instruments, Barcelona, Spain) and activity water (aw) (Aqualab 4 TE—Decagon Devices Inc., Pullman, WA, USA). The moisture (%) was determined on 100 g of meat by oven drying for 24 h at 105 °C [31].

### 2.7. Meat Extract Preparation

Buffalo meat samples of both experimental groups were cut in small pieces and homogenized with three parts of 0.1% (*w*/*w*) formic acid precooled at 4 °C using a Precellys 24 system (Bertin Technologies, Montignyle-Bretonneux, France). The homogenate was centrifuged at 10,000× *g* at 4 °C for 10 min and the clarified supernatant filtered sequentially through a 5 μm and 0.45 μm Millipore filters. Before mass spectrometric analysis, aliquots were filtered through Amicon Ultra 0.5 mL centrifugal filters with 3 kDa molecular weight cutoff.

### 2.8. Betaine and Carnitine Content in Meat Extract

The level of γ-Butyrobetaine, Glycine betaine, δ-Valerobetaine, L-Carnitine, Acetyl-L-carnitine, Propionyl-L-carnitine, and Butyrylcarnitine was determined as previously described [32]. Analysis involved HPLC-ESI-MS/MS tandem mass spectrometry with an Agilent LC-MSD SL quadrupole ion trap and a 1100 series liquid chromatograph (Supelco Discovery C8 column, 250 × 3.0 mm, particle size 5 μm) under isocratic conditions, 0.1% formic acid in water, at flow rate of 100 μL/min. The quantification of each compound involved comparison of the peak area of its most intense MS2 fragment [21,32] with the respective calibration curve built with solutions of standards (Sigma-Aldrich; Milan, Italy). Standard solutions were prepared by serial dilution of standard stock solutions (2 mg/L) with water containing 0.1% formic acid. Linearity was assessed by correlation coefficients (R^2^) > 0.99 for all compounds. Precision and accuracy for all compounds in milk ranged from 95% to 105%.

### 2.9. Microbiological Determination

For microbiological analysis, ten grams of matrix were aseptically weighted from different portions of sample, transferred in a sterile stomacher bag with 90 mL (1:10 (*w*/*w*)) of sterilized Peptone Water (PW, Oxoid, Madrid, Spain), and homogenized for three minutes at 230 rpm in peristaltic homogenizer (BagMixer^®^400 P, Interscience, Saint Nom, France). Tenfold serial dilutions of each homogenate were prepared. Procedures validated by the UNI CEI EN ISO/IEC 17025 European Standard were used to enumerate investigated microbial communities. Specifically, total aerobic bacterial counts (TAB) were counted on plate count agar (PCA; Oxoid, Madrid, Spain) and incubated at 30 °C for 48/72 h (ISO 4833-1:2013); Enterobacteriaceae were enumerated on violet red bile glucose agar (VRBGA), incubated at 37 °C for 24 h (ISO 21528-2:2017); to measure coagulase positive Staphylococcus, Baird Parker medium with RPF supplement (Oxoid, Madrid, Spain) was used and incubated at 37 °C for 24–48 h (ISO/DIS 6888-1); while Mannitol salt-phenol-red agar (MSA; Oxoid, Madrid, Spain) was used for the enumeration of presumptive coagulase-negative staphylococci (CNS), incubated at 30 °C for at least two days; yeasts and molds were plated on Dichloran Rose-Bengal Chloramphenicol Agar (DRBC, Oxoid, Madrid, Spain) incubated at 25 °C for 120/168 h (ISO 21527:2008). All analyses were performed in duplicate. The results were expressed as logarithms of the number of colony forming units (CFU)/g and means and standard error were calculated.

### 2.10. Challenge Test

The challenge test lasted 21 days, storing post maturation meat at 2 °C and 4 °C for their shelf life. To carry out the study, part of buffalo meat samples of the S+/F group were contaminated with L. monocytogenes, and were stored at 2 ± 1 °C and at 4 ± 1 °C, while buffalo untreated meat samples of the same group were used as control. Meat contamination was performed in agreement with the Guideline of the European Reference Laboratory for L. monocytogenes [33].

A suspension of two different strains was used as inoculum, including the reference strain 229 = ATCC7644227M—EURL 12M0B098LM [34]. The preparation of the inoculum was done following the European Reference Laboratory for L. monocytogenes guideline [33] with some modifications (concentration of the inoculum and use of a mix of two L. monocytogenes strains). The challenge test was carried out to assess the growth potential of bacteria in aging buffalo meat. The strains, stored in Microbank™ (Pro-Lab Diagnostics, Round Rock, TX, USA) at −80 °C, were individually revitalized at 37 ± 1 °C for 18 h in BHI broth (Brain Heart Infusion, Sigma-Aldrich Co. LLC, St. Louis, MI, USA). In order to get adapted to the meat storage temperature, new subcultures were set up in BHI broth and incubated at 2 ± 1 °C and 4 ± 1 °C until reaching the stationary phase of each bacterial strain [33,35]. The inoculum suspension was then prepared, consisting of equal parts of liquid cultures of the two selected strains. The mixture was enumerated according to ISO 11290-2:1988/Amd. 1:2004. The suspension was subsequently diluted (saline water) to obtain a concentration of 103 CFU/g in the product.

Enumeration of Listeria monocytogenes was carried out in triplicate on inoculated and control samples by diluting the sample (10–15 g) at 1:5 in peptone saline solution (8.5 g/L NaCl, 1 g/L of peptone) and spreading on Palcam agar, incubated at 37 °C for 48 h. Samples were tasted at different times (D = days), D1, D2, D3, D4, D5, D8, D11, D14, D18, and D21. The results obtained were expressed as Log CFU/g, and were used to calculate the trend of the concentration in contaminated samples.

The calculation of growth potential (δ) of L. monocytogenes was performed according to the guidelines of the EURL Lm. The parameter represents the difference between the logarithmic medians of the counts detected at the end and at the beginning of the challenge test, respectively. Foods able to support L. monocytogenes growth are characterized by δ value greater than 0.5. The growth potential was calculated for each lot, using median values.

### 2.11. Statistical Analyses

Statistical analyses were performed using SPSS (23.0) for Windows 10 (SPSS Inc., Chicago, IL, USA) [36]. All data were presented as the least square mean ± standard error (SEM). To evaluate differences among means, parametric tests such as *t*-test or Tukey’s test were performed for each significant effect (*p* < 0.05). Data on PDA parameters (meat physical-chemical characteristics, functional molecules, challenge test) were analyzed by analysis of variance (ANOVA) for repeated measures with time as the main factor. Buffaloes were tested within treatments. Day of sampling was the repeated measure.

## 3. Results

### 3.1. Effects of Availability of Space on Animal Growth

No differences were recorded for dry matter intake (8.39 ± 0.02 vs. 8.17 ± 0.01 kg in Groups S− and S+ respectively) and daily weight gain during the fattening period (Table 3). The live body weights measured at slaughter were 520 ± 4.7 and 513 ± 7.0 kg in S+ and S− groups, respectively. Moreover, the carcass weight was not different between groups (255.3 ± 9.0 kg on average). The S.E.U.R.O.P. score was R2 class and the average carcass yield was 49% in both groups. However, the stripping rate showed different values between the two groups. The S+ group showed higher percentages of lean mass (62.7% vs. 52.9%; *p* < 0.01) and lower percentages of bone mass (25.8% vs. 38.3%; *p* < 0.01) on total carcass compared with the S− group. No differences were found for the fat mass (10.9% vs. 7.7%) (Table 3).

### 3.2. Effects of Feeding on Animal Growth

Regarding the effects of different feeding regimen, no differences were recorded for the dry mater intake, the daily average weight gain, and live body weights measured at slaughter between two experimental groups during the fattening period (Table 3). Moreover, the carcass weight was not different between groups (272.8 ± 2.9 kg on average). The S.E.U.R.O.P. score was R2 class and the average carcass yield was 51% in both groups. No differences were recorded in terms of percentages of lean mass, fat mass, and bone mass between groups (Table 3).

### 3.3. Effects of Space Allocation and Feeding on Betaines, Carnitine, and Short-Chain Acylcarnitines

Regarding space availability, the content of betaine, carnitine, and short chain acylcarnitines in the meat of S+ group is statistically higher for most of the molecules taken into account (GlyBet, δVB, C3Cnt *p* < 0.01; and Cnt, C2Cnt; *p* < 0.05) compared with the S− group. Only γBB and n-C4Cnt showed no differences between groups (Table 4).

Betaine, carnitine, and short chain acylcarnitines content in S+/F group is statistically higher for most of the molecules taken into account (GlyBet, δVB, C_3_Cnt *p* < 0.01; and Cnt, C_2_Cnt, n-C_4_Cnt; *p* < 0.05) compared with the S+/D group. Only γBB did not change (Table 4).

### 3.4. Effects of Post Dry Aging on Physical-Chemical Characteristics of Meat

At slaughter, animals of S+/F Group had a pH value of 5.6 ± 0.02 and an aw value of 0.98 ± 0.01. The effects of post dry aging on physical-chemical characteristics of meat are reported in Table 5.

### 3.5. Effects of Post Dry Aging on Functional Metabolites of Meat

Regarding functional metabolites, all of them, except for the γ-butyrobetaine, increased during maturation time, up to 60 days (*p* < 0.01; Table 6).

### 3.6. Effects of Post Dry Aging on Microbiological Analysis

No differences were recorded in terms of microbiological analysis during PDA (Table 7).

### 3.7. Challenge Test

*L. monocytogenes* was not found in the control group. Trends of *Listeria* concentration in the inoculated samples are shown in Table 8. As shown in Figure 1, the storage temperature significantly influences (*p* < 0.01) the bacteria survival, and the refrigeration at 2 °C appeared able to control the replicative capacity of this pathogenic microorganism. In particular, at the beginning of experimental period, the group stored at 2 °C showed a significant increase in *Listeria* inoculated until T3 (+0.54 Log CFU/g from D0). Conversely, a gradual decrease in *Listeria* loads occurred during the rest of the trial, registering a δ value of −1.23 at D21.

Different results were collected for the meat group stored at 4 °C. As shown in Table 8, *Listeria* load significantly differed between groups, clearly underlining the bacteria ability to adapt itself to grow under refrigeration conditions, preferring the common temperature of cold rooms and fridges (4 °C). Therefore, a constant increase in *Listeria* inoculated in the product was obtained, showing a gradual growth to a maximum of 5.19 ± 0.02 Log (CFU/g) at D21. The value of δ calculated for this meat group at the end of trial was equal to 3.01.

According to the guidelines of the EURL Lm, food is considered able to support *L. monocytogenes* growth when the δ value is greater than 0.5; therefore, a storage at 2 °C could make the aged meat an ideal matrix for the survival of this pathogenic bacteria.

## 4. Discussion

In the present study, the use of different breeding techniques on buffalo bulls improves the meat concentrations of most of the health-promoting biomolecules examined. Moreover, the use of a PDA system enhances the amount of biomolecules during maturation time without affecting the meat microbiological profile. In the first experiment, S− buffaloes grew similarly to their S+ counterparts without a change in their ingestion rate and no differences were recorded in vivo. However, at the slaughter house, S+ animals showed higher percentages of lean and fat mass on total carcass compared with S− animals, in which the bone mass is predominant on total carcass. It could be hypothesized that a higher availability of space could enhance a better development and differentiation of tissues. In addition, a reduced space allowance may affect the movements needed to stand up or lie down, as lying patterns were restricted by other bulls. A decreased level of lying idle may have detrimental effects on animal welfare and could increase aggression phenomenon [37]. In this experiment, as already experienced by Salzano et al. [24], S− buffaloes may have increased stress levels and elevated blood glucocorticoids, which, in turn, could affect the production of biomolecules. Indeed, the higher amount of biomolecules, recorded in the meat of S+ group, could potentially be explained by a shift in the glycogenolysis/gluconeogenesis ratio to glycogenolysis, as a result of the increased production of carnitine and its precursors. This is because carnitine can influence glucose biosynthesis through the conversion of the available free CoA [38]. The lower content of carnitine and its precursors in the meat of S− group could also be explained by a competition to the manger access and a lower ruminating time [39,40], although, in our study, no differences were observed in DMI. Indeed, a lower ruminating time could affect the synthesis of the biomolecules of interest that are influenced by the persistence of the feed in the reticolo-rumen and through the action of specific bacteria [41].

In the second experiment, buffalo bulls were all reared in 15 m^2^, but they experienced a different diet based on the presence or absence of green forage. In this case, no differences were reported both in vivo, in the ingestion rate, and at the slaughter house measurement. However, the amount of biomolecules in meat was higher in the S+/F group compared with the S+/D group and our data are in agreement with Salzano et al. [25], who demonstrated how a forage-based diet could positively influence the content of biomolecules in buffalo milk. The possible explanation of these results comes from the natural source of carnitine, the Nε-trimethyl lysine (TML), which is particularly abundant in leafy vegetables [32]. Moreover, the inclusion of some leafy vegetables such as clover and ryegrass in animal diet has also produced favorable nutrient profiles in terms of gross composition, macroelements, and trace elements in milk [42,43], and in fatty acid composition and antioxidant content in meat [44,45]. On the contrary, diets based on a total mixed ration are very poor in TML and its amount is minimal [46].

Based on these results, we used the best group to assess the biomolecules content during a PDA maturation process. Our findings demonstrated that a maturation period, in a controlled system (Maturmeat^®^) up to 60 days, did not alter meat pH, aw, and weight loss, nor the microbiological profile, while it enhanced the levels of L-carnitine and its precursor. Among biomolecules, δ-valerobetaine has both antioxidant and anti-inflammatory properties, suppressing pro-inflammatory cytokines (TNF-α, IL-1β, and IL-6) through NAD-dependent deacetylase sirtuin-1 (SIRT1) and SIRT6 modulation [21]. Other recent studies [22,23] reported that δ-valerobetaine also has an antineoplastic effect in human head and neck squamous carcinoma cell lines. This evidence supports the role of these molecules as an important epi-nutrient that should be included in a healthy and balanced diet.

It is worth noting that, after 60 days in the Maturmeat^®^, the functional quality of raw ground buffalo meat has improved and the product could be directly sold in restaurants as a ready-to-eat food (e.g., tartare, carpaccio) that is often requested by consumers. However, these biomolecules have a high thermal stability and could also resist through the digestive tract passage [47], so meat could be cooked without losing its properties.

The inoculation of pathogenic Listeria monocytogenes in aged buffalo meat allowed to simulate possible contamination of foods during processing phases along the whole chain. The results obtained pointed out that the conservation at 2 °C of contaminated aged meat is able to slow down bacteria growth. Di Ciccio et al. [48] reported that L. monocytogenes have acquired abilities to survive and adapt themselves to environments that do not support its growth, developing resistances against low temperatures and a jump in pH. However, growth potential values of the meat group stored at 2 °C were very low, confirming the real potential role of temperature in food safety.

## 5. Conclusions

In conclusion, the present study showed how the increase in area per animal and the inclusion of green fodder could enhance animal production in terms of quality, without affecting eco sustainability. At present, further studies are ongoing to analyze the exact amount of biomolecules precursors in the all components of the animal diet. A better comprehension of the feedstuff nutritional profile would lead to the creation of personalized diet, according to the needs of the single animal. This could be important for all the stages of life, not only for the fattening period, as analyzed in this paper. Moreover, the PDA maturation system could also be modified according to different species or pieces of meat, enhancing the nutraceutical profile of the raw meat.

The quality of dry aging buffalo meat is not only maintained, but improved with the aid of PDA maturation systems, with a creation of a final product that has beneficial properties to human health and wellbeing. Moreover, a high-quality product is strongly requested by final consumers and could be easily sold even at higher prices, guaranteeing greater incomes to breeders and entrepreneurs.

## Figures and Tables

**Figure 1 vetsci-08-00066-f001:**
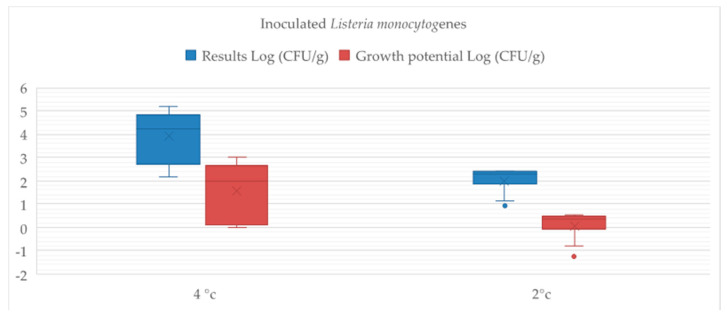
Effect of post dry aging on Listeria growth in inoculated samples of buffalo S+/F meat stored at 2° and 4 °C. Listeria concentration (Log (CFU/g)) and growth potential (Log (CFU/g)) are expressed as means ± SEM for each inoculated group.

**Table 1 vetsci-08-00066-t001:** Feed (kg) and chemical composition (% of dry matter, DM) of the diet for buffalo bulls reared at 10 and 15 m^2^/head.

Feed	kg
Straw	3
Concentrate	7
Water	8
TOTAL	18
MFU/kg DM	**Nutritive Value**
	0.93
TDN (%)	69.4
	**Chemical Composition**
Dry Matter (kg)	8.6
Dry matter (%)	47.7
Moisture (%)	52.3
CP (%)	12.3
EE (%)	3.5
NDF (%)	36.0
ADF (%)	16.0
ADL (%)	5.0
NSC (%)	42.2
Calcium (%)	0.7
Phosphorus (%)	0.4

MFU = meat forage unit; TDN = total digestible nutrients; CP = crude protein; EE = ether extract; NDF = neutral detergent fiber; ADF = acid detergent fiber; ADL = acid detergent lignin; NSC = non-structural carbohydrates.

**Table 2 vetsci-08-00066-t002:** Feed (kg) and chemical composition (% of dry matter, DM) of the diet in buffaloes with (S+/F) or without (S+/D) rye grass inclusion.

Feed	Diet (kg)	
	S+/D	S+/F
Rye grass	-	12
Straw	3	2
Concentrate	7	5
Calcium carbonate	-	0.1
Water	8	-
TOTAL	18.0	19.1
	**Nutritive Value**	
MFU/kg DM	0.93	0.93
TDN (%)	69.4	69.4
	**Chemical Composition**	
Dry Matter (kg)	8.6	8.6
Dry matter (%)	47.7	44.7
CP (%)	12.3	12.2
EE (%)	3.5	4.0
NDF (%)	36.0	36.1
ADF (%)	16.0	19.4
ADL (%)	5.0	5.4
NSC (%)	42.2	41.7
Ca (%)	0.7	0.7
P (%)	0.4	0.4

MFU = meat forage unit; TDN = total digestible nutrients; CP = crude protein; EE = ether extract; NDF = neutral detergent fiber; ADF = acid detergent fiber; ADL = acid detergent lignin; NSC = non-structural carbohydrates.

**Table 3 vetsci-08-00066-t003:** Effects of availability of space and different feeding regimen on animal growth, daily average gain, dry matter intake, live body weight at slaughterhouse, and carcass weight and yield of buffalo meat in animal bred at 10 m^2^ (S−), at 15 m^2^ (S+), and fed total mixed ratio (S+/D) or green forage (S+/F). Values are expressed in percentages or as the mean ± ES.

Group	Daily Average Gain (g)	Dry Matter Intake (%)	Live Body Weight (kg)	Carcass Weight (kg)	Lean Mass (%)	Bone Mass (%)	Fat Mass (%)
S+	720 ± 16.7	1.72	520 ± 4.7	257.7 ± 8.1	62.7 ^A^	25.8 ^A^	10.9 ^a^
S−	675 ± 15.9	1.75	513 ± 7.0	252.8 ± 9.9	52.9 ^B^	38.1 ^B^	8.7 ^a^
S+/D	838 ± 9.2	1.73	533 ± 7.0	271.8 ± 3.4	57.8	22.9 ^A^	18.9 ^b^
S+/F	849 ± 12.0	1.70	540 ± 4.7	273.7 ± 2.4	58.9	22.5 ^A^	18.5 ^b^

^a,b^ values within columns differ *p* < 0.05; ^A,B^ values within columns differ *p* < 0.01.

**Table 4 vetsci-08-00066-t004:** Functional metabolites in buffalo meat in animal bred at 10 m^2^ (S−), at 15 m^2^ (S+), and fed total mixed ratio (S+/D) or green forage (S+/F). Metabolites’ concentration (mg/Kg) is expressed as the mean ± ES. Statistical analysis was performed using *t*-test. Glycine betaine, GlyBet; γ-Butyrobetaine, γBB; δ-Valerobetaine, δVB; L-Carnitine, Cnt; Acetylcarnitine (C_2_Cnt); Propionylcarnitine (C_3_Cnt); Butyrylcarnitine (n-C_4_Cnt).

	GlyBet	γBB	δVB	Cnt	C2Cnt	C3Cnt	n-C4Cnt
S−	54.50 ± 2.52 ^A^	13.27 ± 0.54	44.41 ± 1.19 ^A^	271.67 ± 5.40 ^a^	236.38 ± 11.24 ^a^	26.26 ± 2.3 ^A^	134.38 ± 4.02
S+	66.57 ± 1.13 ^B^	13.20 ± 0.64	55.73 ± 1.81 ^B^	285.54 ± 3.22 ^b^	294.10 ± 13.65 ^b^	37.21 ± 1.66 ^B^	158.06± 4.76
S+/D	68.90 ± 0.60 ^A^	14.54 ± 0.55	56.83 ± 1.8 ^A^	293.14 ± 3.43 ^A^	295.19 ± 4.02 ^a^	36.34 ± 1.22 ^a^	162.59 ± 4.51 ^a^
S+/F	79.67 ± 1.45 ^B^	14.29 ± 1.14	71.13 ± 1.37 ^B^	317.40 ± 5.06 ^B^	330.16 ± 5.74 ^b^	42.89 ± 1.31 ^b^	174.57 ± 1.56 ^b^

^a,b^ values within columns differ *p* < 0.05; ^A,B^ values within columns differ *p* < 0.01.

**Table 5 vetsci-08-00066-t005:** Activity water (aw), pH, and weight loss during meat maturation time comes from animals fed with green forage and bred at low density (S+/F) in Longissimus muscle at T0, 30 days (T1), and 60 days (T2) after maturation.

Group	aw			pH			Weight Loss (kg)		
	T0	T1	T2	T0	T1	T2	Weight T0	Weight Loss T1	Weight Loss T2
S+/F	0.98 ± 0.00	0.98 ± 0.00	0.98 ± 0.00	5.5 ± 0.02	5.7 ± 0.01	5.7 ± 0.03	18.1 ± 0.9	1.4 ± 0.0	2.1 ± 0.0

**Table 6 vetsci-08-00066-t006:** Functional metabolites in meat of buffalo bulls fed with green forages and rear at low density conditions at T0, 30 days (T1), and 60 days (T2) after maturation. Metabolites’ concentration (mg/Kg) is expressed as the mean ± ES.

Metabolites	T0	T1	T2
Glycine betaine	80.6 ± 3.1 ^B^	118.3 ± 8.3 ^b^	160.3 ± 17.5 ^A^
γ-Butyrobetaine	15.2 ± 0.8	17.2 ± 0.4	18.9 ± 0.7
δ-Valerobetaine	66.5 ± 4.8 ^B^	90.7 ± 3.1 ^b^	129.0 ± 5.5 ^A^
L-Carnitine	354.7 ± 8.0 ^B^	424.0 ± 4.3 ^b^	495.1 ± 9.3 ^A^
Acetylcarnitine	328.0 ± 7.0 ^B^	377.1 ± 3.1 ^b^	449.2 ± 22.0 ^A^
Propionylcarnitine	41.2 ± 1.1 ^B^	50.0 ± 0.8 ^b^	57.2 ± 2.7 ^A^
Butyrylcarnitine	171.8 ± 3.4 ^B^	193.0 ± 4.0 ^b^	216.2 ± 6.1 ^A^
Choline	16.1 ± 1.9 ^B^	22.7 ± 0.7 ^B^	43.8 ± 2.4 ^A^
Glycerophosphocholine	206.8 ± 9.5 ^B^	259.0 ± 8.0 ^B^	379.7 ± 17.8 ^A^

^A,B^ values within row differ *p* < 0.01; ^A,b^ values within row differ *p* < 0.05.

**Table 7 vetsci-08-00066-t007:** Effect of post dry aging on microbiological profile measured as Log (CFU/g) of buffalo meat fed ray grass and bred at low density (S+/F) in Longissimus muscle at T0, 30 days (T1), and 60 days (T2) after maturation.

	Mesophilic Microbe Content	Enterobacteriaceae	Coagulase-Positive Staphylococci	Coagulase-Negative Staphylococci	Yeast	Mold
S+/F T0	4.70 ± 0.31	1.35 ± 0.00	4.21 ± 0.26	4.74 ± 0.65	3.01 ± 0.34	2.05 ± 0.10
S+/F T1	4.72 ± 0.52	1.40 ± 0.18	4.17 ± 0.58	4.43 ± 0.63	2.85 ± 0.67	2.05 ± 0.13
S+/F T2	4.76 ± 0.29	1.30 ± 0.00	4.21 ± 0.65	4.74 ± 0.32	2.97 ± 0.65	2.76 ± 0.40

**Table 8 vetsci-08-00066-t008:** Effect of post dry aging on *Listeria* growth in inoculated samples of buffalo S+/F meat stored at 2° and 4 °C. *Listeria* concentration (Log (CFU/g)) is expressed as the mean ± SEM for each inoculated group and at each sampling time.

Day	Log (CFU/g)				Growth Potential (Log (CFU/g))	
	2 °C	*p*	4 °C	*p*	2 °C	4 °C
	Mean ± SEM		Mean ± SEM		δ	δ
D0	1.90 ± 0.03 ^A^		2.17 ± 0.08 ^B^		-	-
D1	2.39 ± 0.02	***	2.44 ± 0.17	NS	0.49	0.26
D2	2.44 ± 0.01 ^A^	*	2.22 ± 0.05 ^B^	NS	0.54	0.05
D3	2.41 ± 0.03 ^A^	NS	3.31 ± 0.03 ^B^	***	0.51	1.14
D4	2.26 ± 0.10 ^A^	NS	4.11 ± 0.02 ^B^	***	0.36	1.94
D5	2.32 ± 0.08 ^A^	NS	4.21 ± 0.03 ^B^	*	0.42	2.04
D8	2.23 ± 0.17 ^A^	NS	4.29 ± 0.06 ^B^	NS	0.33	2.11
D11	2.06 ± 0.10 ^A^	NS	4.44 ± 0.06 ^B^	NS	0.15	2.27
D14	1.84 ± 0.12 ^A^	NS	4.92 ± 0.04 ^B^	***	−0.06	2.75
D18	1.10 ± 0.10 ^A^	***	5.16 ± 0.04 ^B^	***	−0.80	2.99
D21	0.67 ± 0.33 ^A^	***	5.19 ± 0.02 ^B^	NS	−1.23	3.01

^A,B^ values within row differ *p* < 0.01, ***** and *** values within column differ *p* < 0.05, *p* < 0.01, and *p* < 0.001, respectively. Statistical analyses were carried out to evaluate *Listeria* growth trend between two consecutive sampling times.

## Data Availability

Not applicable.

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
