# Peer review of "Effect of Breeding Techniques and Prolonged Post Dry Aging Maturation Process on Biomolecule Levels in Raw Buffalo Meat"

_vetsci, 2021, doi:10.3390/vetsci8040066_

Round 1

Reviewer 1 Report

MANUSCRIPT:

“TITLE: EFFECT OF PRODUCTION SYSTEMS AND PROLONGED POST DRY AGING 2 MATURATION PROCESS ON BIOMOLECULES LEVELS IN RAW BUFFALO MEAT”

Review

Considerations

The study evaluated the spacing effect (10 or 15m2 per animal, S- or S +, respectively) and the inclusion or not of forage in the diet (S + F or S + D) on buffalo production. The work presents interesting results from a scientific point of view.

However, a lot of information needs to be inserted in order for the work to have merit for publication and to be replicated by other researchers.

Lines 1-3: The evaluated production system is classified as an intensive production system (10 and 15m2 per animal). Different production systems were not evaluated, but the effect of spacing and nutrition (Breeding techniques). With that, a readjustment in the title is interesting.

Line 16: Assessing animal welfare does not seem to be the main objective of the study, although there may be a correlation, the objective was to assess production density and diet.

Line 21: What green forage

Line 86: animal welfare was not one of the direct parameters evaluated, but the spacing that can influence both welfare and productive characteristics.

Line 106: Information on feeding times, calculation of intake rate, leftovers (ad libitum) - Important information for replicating this study.

Table 1

What was the rate of ingestion of the animals (% BW) there were differences in consumption between the two spacings?

What is the purpose of entering the water consumption data ... Was this nutrient provided ad libitum?

Analysis of ether extract content, total digestible nutrients in the diet? Isoenergetics?

Kg of dry matter of water?

Did the animals consume 8.6 kg of dry fed?

Water consumption of these animals can be around 6% live weight consumption (30-40l vs. 8l)

Line 119 - Rye: Secale cereale

Line 119 - Energy values? Etheric Extract Content? Total Digestible Nutrients

Table 2

Do you hear a limitation of water consumption for diets with forage?

Analysis of the ether extract content of the diet? NDT data is missing

Results

Line 246-248: There was no difference between the values, so within the space the numbers are similar (MEAN 697, X ± X, XX)

Line 253: There was no difference between the values, so having an average value

Line 255: Has a precursor component of these biomolecules of interest in the carcass been evaluated in the diet? If yes, enter it in the chemical composition tables

Lines 244-253: These carcass weight data and carcass characteristics would be interesting to present in tables. Mainly to increase the citations of the work in a possible revision work.

Lines 270-275: There was no difference between the values ​​so the average value of the treatments must be entered

Line 370 Has there been a change in the rate of ingestion of the different density treatments?

Lines 385-387 Diet intake data (Suggestions in Table 1) would be interesting to assist in this statement.

Lines 393-400: It would be interesting to bring aspects of the diet components, effect on the ruminal biohydrogenation process and deposition in the carcass, even being able to use classic beef cattle works.

Line 393 An evaluation of these components was carried out in the diet with and without green forage

Conclusion

Line 427: The question of well-being is an assumption ... Bring components evaluated in the project (spacing and diet)

The increase in area per animal and the inclusion of green fodder provided.

Author Response

Reviewer 1

Considerations

The study evaluated the spacing effect (10 or 15m2 per animal, S- or S +, respectively) and the inclusion or not of forage in the diet (S + F or S + D) on buffalo production. The work presents interesting results from a scientific point of view.

However, a lot of information needs to be inserted in order for the work to have merit for publication and to be replicated by other researchers.

Reply: Additional information for clarity has been included throughout the manuscript as requested by the reviewer.

 Lines 1-3: The evaluated production system is classified as an intensive production system (10 and 15m2 per animal). Different production systems were not evaluated, but the effect of spacing and nutrition (Breeding techniques). With that, a readjustment in the title is interesting.

Reply: Corrected as suggested by reviewer 1 (lines 1-3 of the revised manuscript).

Line 16: Assessing animal welfare does not seem to be the main objective of the study, although there may be a correlation, the objective was to assess production density and diet.

Reply: Corrected as suggested by reviewer 1 (lines 14-16 of the revised manuscript).

Line 21: What green forage

Reply: Added as suggested by reviewer 1 (lines 18-19 of the revised manuscript).

Line 86: animal welfare was not one of the direct parameters evaluated, but the spacing that can influence both welfare and productive characteristics.

Reply: Corrected as suggested by reviewer 1 (lines 91-94 of the revised manuscript).

Line 106: Information on feeding times, calculation of intake rate, leftovers (ad libitum) - Important information for replicating this study.

Reply: The lacking information were now added, as suggested by reviewer 1 (lines 110-114 and 131-136 of the revised manuscript).

Table 1

What was the rate of ingestion of the animals (% BW) there were differences in consumption between the two spacings?

Reply: No differences were recorded in consumption between the two Groups and the rate of the ingestion of the animal was now added in Table 3.

What is the purpose of entering the water consumption data ... Was this nutrient provided ad libitum?

Reply: This data regards the amount of water in the total mixed ration to ensure a correct mixture of the feedstuffs, nor the total amount of water consumed daily by the animals.

Analysis of ether extract content, total digestible nutrients in the diet? Isoenergetics?

Reply: The diet reported in Table 1 is the same for both Groups. The two diets reported in Table 2 were isoenergetics and isoproteic for both Groups. Data of Ether extract and TDN were now reported in both Tables.

Kg of dry matter of water?

Reply: In Table 1 we reported the kg of water used in the total mixed ration to ensure a correct mixture of the feedstuffs of dry matter and also the percentage of moisture.

Did the animals consume 8.6 kg of dry fed?

Reply: The amount of dry matter (DM), expressed as percentage of body weight has been reported in Table 3. In any case, the DM consumed by the animals of S- Group was 8.39 kg, by the S+ Group 8.17 kg, by S+/D Group was 8.57 kg and by the S+/F Group was 8.53 kg.

Water consumption of these animals can be around 6% live weight consumption (30-40l vs. 8l)

Reply: This is the amount of water present in the diet, the animals have also free access to water but we did not evaluate the total amount of water intake.

Line 119 - Rye: Secale cereale

Reply: It was Lolium multiflorum. This information was added to the revised manuscript (line 130 of the revised manuscript).

Line 119 - Energy values? Etheric Extract Content? Total Digestible Nutrients

Reply: The two diets were isoenergetics (MFU/DM = 0.93) and the data of Ether extract and TDN were now reported.

Table 2

Do you hear a limitation of water consumption for diets with forage?

Reply: Water consumption was not calculated.

Analysis of the ether extract content of the diet? NDT data is missing

Reply: Data were now reported.

 Results

Line 246-248: There was no difference between the values, so within the space the numbers are similar (MEAN 697, X ± X, XX)

Reply: We decided to put these numbers in Table 3.

Line 253: There was no difference between the values, so having an average value

Reply: Corrected as suggested by reviewer 1 (lines 268-269 of the revised manuscript).

Line 255: Has a precursor component of these biomolecules of interest in the carcass been evaluated in the diet? If yes, enter it in the chemical composition tables

Reply: One of the precursors of these biomolecules is Nε-trimethyl lysine, scarcely present in protein sources (soybean protein, casein and wheat gluten), is highly present in leafy vegetables, in particular alfalfa (Servillo et al 2014). At present, unfortunately, no data were available for the amount of TML in carcass.

Servillo, L., Giovane, A., Cautela, D., Castaldo, D., & Balestrieri, M.L. ,2014. Where does N(ε)-trimethyllysine for the carnitine biosynthesis in mammals come from? PLoS One, 9, e84589).

Lines 244-253: These carcass weight data and carcass characteristics would be interesting to present in tables. Mainly to increase the citations of the work in a possible revision work.

Reply: Corrected as suggested by reviewer 1, now everything is reported in Table 3.

Lines 270-275: There was no difference between the values ​​so the average value of the treatments must be entered

Reply: Corrected as suggested by reviewer 1 (lines 279-280 of the revised manuscript).

Line 370 Has there been a change in the rate of ingestion of the different density treatments?

Reply: We thank the reviewer for the interesting question. No, differences were observed for the dry matter intake within groups. We added this information at lines 390-391 and 412-413 of the revised manuscript.

Lines 385-387 Diet intake data (Suggestions in Table 1) would be interesting to assist in this statement.

Reply: We thank the reviewer that allow us to improve the discussion section with this suggestion. The diet intake data were similar between Groups so probably no competition to the manger or lower ruminating time were observed in our study. We add this statement at lines 446-449 of the revised manuscript.

Lines 393-400: It would be interesting to bring aspects of the diet components, effect on the ruminal biohydrogenation process and deposition in the carcass, even being able to use classic beef cattle works.

Reply: We acknowledge the reviewer for the very interesting question. Unfortunately, we did not measure or focus our attention on the effect on the ruminal biohydrogenation process and deposition in the carcass. It is one of the objectives of our future studies.

Line 393 An evaluation of these components was carried out in the diet with and without green forage?

Reply: The TML content is known to be high in alfalfa (Servillo et al 2014). In the present study, the content of TML in the diet has not been evaluated as future studies aimed at evaluating the impact of the diet (with and without green forage) on ruminal metabolism would further sustain the impact of forage-based diet on the content of biomolecules.

Servillo, L., Giovane, A., Cautela, D., Castaldo, D., & Balestrieri, M.L. ,2014. Where does N(ε)-trimethyllysine for the carnitine biosynthesis in mammals come from? PLoS One, 9, e84589)

Regarding other components in meat such as fatty acid composition or an evaluation of meat colour and tenderness, we have published a recent paper (Marrone et al., 2020) on the effect of a combined used of green forages and a post dry aging maturation period. We specified this at lines 78-81 of the revised manuscript. The present study was more focused on the functional quality of buffalo meat.

Marrone, R.; Salzano, A.; Di Francia, A.; Vollano, L.; Di Matteo, R.; Balestrieri, A.; Anastasio, A.; Barone, C.M.A. Effects of Feeding and Maturation System on Qualitative Characteristics of Buffalo Meat (Bubalus bubalis). Animals (Basel) 2020, 10, 899. doi: 10.3390/ani10050899.

Conclusion

Line 427: The question of well-being is an assumption ... Bring components evaluated in the project (spacing and diet)

Reply: Corrected as suggested by reviewer 1 (lines 451-453 of the revised manuscript).

The increase in area per animal and the inclusion of green fodder provided.

Reply: Corrected as suggested by reviewer 1 (lines 451-453 of the revised manuscript).

Reviewer 2 Report

Title: Effect of production systems and prolonged post dry aging maturation process on biomolecules levels in raw buffalo meat

The manuscript is well-written, with an interesting experimental design and important contributions to this field, mainly for the ruminants meat industry sector.

Some clarifications should be needed:

Why the pH value was not recorded at day 0?  Maturation is needed but an initial point of view would have been interesting.

In general in Tables 3, 4, 6,  why are used capital and lower case superscripts? A, B and a, b are indicated for same?

In Table 8 a, b letters are not used in Table so please remove from the footnote legend.

Tables 3 and 4 could be put together in just one Table. In this sense information from productive parameters (Effect of availability of space on animal growth and Effect of feeding on animal growth) could be added in a new Table.

Author Response

Reviewer 2

The manuscript is well-written, with an interesting experimental design and important contributions to this field, mainly for the ruminants meat industry sector.

Some clarifications should be needed:

Why the pH value was not recorded at day 0?  Maturation is needed but an initial point of view would have been interesting.

Reply: We thank the reviewer 2 for this interesting question. Our aim was focused on evaluate different in pH and activity water (aw) during maturation into the Maturmeat, however we agree with the reviewer and data regarding pH and aw measured at slaughtered were added at lines 326-328 of the revised manuscript.

In general in Tables 3, 4, 6, why are used capital and lower case superscripts? A, B and a, b are indicated for same?

Reply: The lower case indicated a statistical difference between Groups with a P value of <0.05, while the capital case indicated a stronger statistical difference between Groups with a P value of <0.01.

In Table 8 a, b letters are not used in Table so please remove from the footnote legend.

Reply: We thank the reviewer that allow us to see a typo. We removed the letters in Table legend.

Tables 3 and 4 could be put together in just one Table. In this sense information from productive parameters (Effect of availability of space on animal growth and Effect of feeding on animal growth) could be added in a new Table.

Reply: Corrected as suggested by reviewer 2 (please see the new Table 4).

Reviewer 3 Report

The manuscript entitled “Effect of production systems and prolonged post dry aging 2 maturation process on biomolecules levels in raw buffalo meat” presents an issue associated with the important public health problem.

Abstract:

  • The aim of the study is “to evaluate/ analysis” not “understand” – please rephrase Line 25 – “microbiological profile” – please be more precise
  • A good abstract should include a problem statement, background, methodology, key finding and a conclusion, which assist the reader to understand the study. 
  • Conclusion must be focused more on the findings
  •  

Introduction section:

  • Line 37-39 – is irrelevant – please remove
  • Line 45 – “this makes buffalo meat particularly suitable for people with cardiovascular risk profile” this sentence is unacceptable and misleading. All recommendationa indicated that the meat and meat products should be reduced and such information could be misleading for the audience. The conclusion from the study is “Consumption of buffalo meat seems to be associated with several beneficial effects on cardiovascular risk profile” – “seems to be” is a crucial fragment!
  • Line 46 – please the statement of WHO or AICR https://www.who.int/news-room/q-a-detail/cancer-carcinogenicity-of-the-consumption-of-red-meat-and-processed-meat; https://www.wcrf.org/dietandcancer/recommendations/limit-red-processed-meat
  • In this section should presented the information associated with the factors influenced the meat quality. This section should be briefly presented – what do we know and what is the background for this study. Some detailed information about other studies are necessary. The good background should present the history of problem, the current knowledge and scientific "gap", and then authors should present how their study could fill this gap to justify the study.
  •  

Materials and methods section:

  • Was the normality of distribution tested? The information about it should be added and authors should be consequent. If data have normal distribution, they should be treated as such, if not, nonparametric tests should be applied. Please specify it.

Results and Discussion section:

  • Tables – all abbreviation must be explained; please explained the time of experiment in legend T1/ T2 / T3 (table 5)
  • Why sensory analysis (and other physicochemical analysis like TPA) was not conducted? Please specify it and justify.
  • Authors should in their discussion include 3 areas: (1) compare gathered data with the results by other authors, (2) formulate implications of the results of their study and studies by other authors, (3) formulate the future areas which should be studied.
  • Authors should present here and discuss the limitations of their study.

Author Response

Reviewer 3

Abstract:

The aim of the study is “to evaluate/ analysis” not “understand” – please rephrase Line 25 – “microbiological profile” – please be more precise

Reply: Corrected as suggested by reviewer 3 (lines 14-16 and 24-26 of the revised manuscript).

A good abstract should include a problem statement, background, methodology, key finding and a conclusion, which assist the reader to understand the study. 

Reply: We tried to modify the abstract according to the reviewer’s suggestions; the 200 maximum word limitation restricts the amount of information that can be included.

Conclusion must be focused more on the findings 

Reply: Corrected as suggested by reviewer 3 (lines 26-28 of the revised manuscript).

Introduction section:

Line 37-39 – is irrelevant – please remove

Reply: Removed as suggested by reviewer 3.

Line 45 – “this makes buffalo meat particularly suitable for people with cardiovascular risk profile” this sentence is unacceptable and misleading. All recommendations indicated that the meat and meat products should be reduced and such information could be misleading for the audience. The conclusion from the study is “Consumption of buffalo meat seems to be associated with several beneficial effects on cardiovascular risk profile” – “seems to be” is a crucial fragment!

Reply: The wording has been revised based on the comments of the reviewer (lines 43-45 of the revised manuscript).

Line 46 – please the statement of WHO or AICR https://www.who.int/news-room/q-a-detail/cancer-carcinogenicity-of-the-consumption-of-red-meat-and-processed-meat; https://www.wcrf.org/dietandcancer/recommendations/limit-red-processed-meat

Reply: Thank you for these websites. We are aware of the risks that an unbalanced human diet, full of red or processed red meat could provide to the consumers. Notwithstanding, red meat is rich in proteins, micro and macro-elements and also functional molecules that provide a good nutrient profile. However, we would agree that it is the decision of individual consumers whether they include red meat in their diet. The literature is still actively debating this topic and we believe that the present manuscript contributes important information to the debate.

In this section should presented the information associated with the factors influenced the meat quality. This section should be briefly presented – what do we know and what is the background for this study. Some detailed information about other studies are necessary. The good background should present the history of problem, the current knowledge and scientific "gap", and then authors should present how their study could fill this gap to justify the study.

Reply: We thank the reviewer 3 that allow us to improve the Introduction section with this good suggestion. From our point of view, the majority of the studies on meat quality are related to the nutritional profile, while we wanted to focus reader’s attention on the importance of the nutraceutical profile, evaluating some important functional molecules, and how we can modulate their levels changing breeding techniques. However, we add more background at lines 52-57 of the revised manuscript.

Materials and methods section:

Was the normality of distribution tested? The information about it should be added and authors should be consequent. If data have normal distribution, they should be treated as such, if not, nonparametric tests should be applied. Please specify it.

Reply: Data have normal distribution and parametric tests were applied (lines 258-259 of the revised manuscript).

 Results and Discussion section:

Tables – all abbreviation must be explained; please explained the time of experiment in legend T1/ T2 / T3 (table 5)

Reply: Corrected as suggested by reviewer 3 (lines 330-332 of the revised manuscript).

Why sensory analysis (and other physicochemical analysis like TPA) was not conducted? Please specify it and justify.

Reply: We did not perform this kind of analysis because another study (Marrone et al., 2020) have already demonstrated that a combined used of green forages and a PDA maturation period could improve both tenderness and colour in buffalo meat. We specified this at lines 78-81 of the revised manuscript. Our study was more focused on the functional quality of buffalo meat.

Marrone, R.; Salzano, A.; Di Francia, A.; Vollano, L.; Di Matteo, R.; Balestrieri, A.; Anastasio, A.; Barone, C.M.A. Effects of Feeding and Maturation System on Qualitative Characteristics of Buffalo Meat (Bubalus bubalis). Animals (Basel) 2020, 10, 899. doi: 10.3390/ani10050899.

Authors should in their discussion include 3 areas: (1) compare gathered data with the results by other authors, (2) formulate implications of the results of their study and studies by other authors, (3) formulate the future areas which should be studied. Authors should present here and discuss the limitations of their study.

Reply: We acknowledge the reviewer 3 that help us improving our manuscript. Limitations and future areas of the study are now added at lines 453-459 of the revised manuscript.

Round 2

Reviewer 1 Report

Reviewer 1

Considerations:

OK

  • Lines 1-3: Title reformulation: ok
  • Line 15: Replacement of animal welfare assessment to evaluate the influence of different breeding techniques: ok
  • Line 18-19: Described the green forage used ok .... Scientific name since it is not in the title or in the keywords
  • Line 91-94: accept.
  • Line 110-114 and 131-136: Information on feeding times, calculation of intake rate, leftovers (ad libitum) - Important information for replicating this study.ok

Table 1

- What was the rate of ingestion of the animals (% BW) there were differences in consumption between the two spacings ?: great

- What is the purpose of entering the water consumption data ... Was this nutrient provided ad libitum? OK

- Analysis of ether extract content, total digestible nutrients in the diet? Isoenergetics? TDN and EE data added: ok

- Kg of dry matter of water? OK

- Did the animals consume 8.6 kg of dry fed? Understood

- Water consumption of these animals can be around 6% live weight consumption (30-40l vs. 8l) ok

  • Line 130 - Lolium multiflorum ok
  • Line 119 - Energy values? Etheric Extract Content? Total Digestible Nutrients: Inserted in the table ok

Table 2

Do you hear a limitation of water consumption for diets with forage? Understood

Analysis of the ether extract content of the diet? NDT data is missing Inserted ok

 Results

  • Line 246-248: There was no difference between the values, so within the space the numbers are similar (MEAN 697, X ± X, XX)

I suggest following a pattern when there is no difference between the Lines 265 and 267 treatments

Do as done line 280 (272.8 ± 2.9 kg on average)

  • Line 253: There was no difference between the values, so having an average value ok
  • Line 255: Has a precursor component of these biomolecules of interest in the carcass been evaluated in the diet? If yes, enter it in the chemical composition tablesAt present, unfortunately, no data were available for the amount of TML in OK Next study
  • Lines 244-253: These carcass weight data and carcass characteristics would be interesting to present in tables. Mainly to increase the citations of the work in a possible revision work. Table 3 OK
  • Lines 279-280: There was no difference between the values ​​so the average value of the treatments must be entered ok
  • Line 370 Has there been a change in the rate of ingestion of the different density treatments? Perfect
  • Lines 446-449 Diet intake data (Suggestions in Table 1) would be interesting to assist in this statement. Correct
  • Lines 393-400: It would be interesting to bring aspects of the diet components, effect on the ruminal biohydrogenation process and deposition in the carcass, even being able to use classic beef cattle works.

Reply: We acknowledge the reviewer for the very interesting question. Unfortunately, we did not measure or focus our attention on the effect on the ruminal biohydrogenation process and deposition in the carcass. It is one of the objectives of our future studies.

Ok. Next

  • Line 393 An evaluation of these components was carried out in the diet with and without green forage?

Reply: The TML content is known to be high in alfalfa (Servillo et al 2014). In the present study, the content of TML in the diet has not been evaluated as future studies aimed at evaluating the impact of the diet (with and without green forage) on ruminal metabolism would further sustain the impact of forage-based diet on the content of biomolecules.

Servillo, L., Giovane, A., Cautela, D., Castaldo, D., & Balestrieri, M.L. ,2014. Where does N(ε)-trimethyllysine for the carnitine biosynthesis in mammals come from? PLoS One, 9, e84589)

Regarding other components in meat such as fatty acid composition or an evaluation of meat colour and tenderness, we have published a recent paper (Marrone et al., 2020) on the effect of a combined used of green forages and a post dry aging maturation period. We specified this at lines 78-81 of the revised manuscript. The present study was more focused on the functional quality of buffalo meat.

Marrone, R.; Salzano, A.; Di Francia, A.; Vollano, L.; Di Matteo, R.; Balestrieri, A.; Anastasio, A.; Barone, C.M.A. Effects of Feeding and Maturation System on Qualitative Characteristics of Buffalo Meat (Bubalus bubalis). Animals (Basel) 2020, 10, 899. doi: 10.3390/ani10050899.

OK, next study

Conclusion

  • Line 427: The question of well-being is an assumption ... Bring components evaluated in the project (spacing and diet)

Reply: Corrected as suggested by reviewer 1 (lines 451-453 of the revised manuscript).

ok

  • The increase in area per animal and the inclusion of green fodder

Reply: Corrected as suggested by reviewer 1 (lines 451-453 of the revised manuscript).

ok

Reviewer 3 Report

I appreciate the great efforts that the authors have made in response to my questions and concerns. However, there are some typos that should be corrected:

line 265 - "8,39±0.02" it should be "8.39±0.02"

line 327 - "and a aw" it should be "and an aw"

line 461 - "The quality is not only maintained" it should be "The quality of buffalo meat is not only maintained" or "The quality of dry aging buffalo meat is not only maintained"